# Impact of Increased Penetration of Low-Carbon Technologies on Cable Lifetime Estimations

**Xu Jiang** [1] , **Edward Corr** [1] , **Bruce Stephen** [2,*] and **Brian G. Stewart** [2]

1 Power Network Demonstration Centre, University of Strathclyde, Cumbernauld G68 0EF, UK; x.jiang@strath.ac.uk (X.J.); edward.corr@strath.ac.uk (E.C.)
2 Institute of Energy and Environment, Royal College Building, University of Strathclyde, Glasgow G1 1XW, UK; brian.stewart.100@strath.ac.uk
* Correspondence: bruce.stephen@strath.ac.uk

**Abstract:** Cables are the largest assets by volume on power distribution networks and the assets with the least health information routinely gathered. Projections over the next 8 years suggest increased penetration of low-carbon technology (LCT) at the distribution level with higher loads resulting from electric vehicle (EV) and heat pump uptake. Over this period, increased cable loading will directly influence their lifetimes and may mean that existing asset management practices need to be revised to understand the specific impact on end-of-life assessment. Accordingly, this paper uses a physics-based thermal lifetime model based on projected uptake trends for LCTs to evaluate the impact on distribution cable lifetime. Two case studies are presented considering portions of network and the ultimate impact on asset life over the next decade. Two commonly used cables are considered to quantify the lifetime reduction caused by LCT for asset fleets. The paper shows that the projected uptake of EVs and heat pumps will have a detrimental effect on cable life with a 30% reduction in cable lifetime possible.

**Keywords:** electric vehicle; heat pump; cable; lifetime assessment; asset management





## 1. Introduction

As government and international bodies focus more on the impact of climate change, low-carbon technologies (LCTs) will become more prominent on power networks. However, as LCT penetration increases, the cable loading regimes will change significantly. For example, in the My Electric Avenue project [1], the British utility SSEN found that approximately one-third of LV networks in the UK need to be upgraded, when 40–70% customers have an EV, and this level of uptake is expected to occur by 2030. These upgrades would result in significant capital expenditure [2] and customer connection disruptions to enable reinforcement works and possible delays in the connection of LCTs. The UK government also endeavors to meet the target of net zero emissions across the economy by 2050 [1]. A key element of this will be switching over 20,000 homes per week from 2025 to 2050 to a low-carbon heat source [3]. Currently, 85% of UK households use natural gas to heat their homes [3]; this is one of the main contributors of the UK's carbon emissions. Heat pumps are one of the most promising solutions, and there is likely to be a targeted replacement strategy for households that are not on the gas network. In addition to the UK, other countries have also set up net-zero targets; for example, the European Union has agreed make climate neutral by 2050 [4]. To date, studies have focused on deferring reinforcement of the power network infrastructure [1,5–7] and the influence of LCT on the power flow [8,9]. No studies have discussed the influence of LCTs on the life of installed power cables, despite this being critical to both.

The lifetime estimation for power cables is important, as they are not visible and represent a large proportion of installed assets for network operators. A Remaining Useful

Life (RUL) predictor for assets can be classified into two potential approaches: data-driven and model-based. The data-driven approach normally applies machine learning algorithms on archived data, such as historical failure data, voltage, and current, to predict RUL. Common models are the Weibull model [10], inverse power law model [11], and Crow–AMSAA model [10]. The data-driven method is focused on the overall trends for a type of cable rather than an individual unit. The availability of run-to-failure and diagnostic data is sometimes limited in this case. Model-based prediction has also been widely investigated to date. Work in Reference [12] used the temperature, humidity, and voltage to calculate the material stress via a cable insulation model, and then the predicted lifetime was estimated on the stress value. Elsewhere, Reference [13] used load, ambient temperature, environmental parameters, and a thermal model of degradation to estimate cable lifetime. In addition to this, recent research has started to combine statistical analysis into model-based predictions. Work in Reference [14] develops a model-based case which used a state-space lifetime equation and a statistical uncertainty model to predict the RUL of cables by using thermal and load data.

The relevant literature indicates that the lifetime of power cables would be significantly affected by the presence of emerging LCTs. A lifetime prediction model is required to help asset managers target which cables need to be reinforced. This approach will mitigate failures, rather than invoke blanket replacement programs, which would represent a significant cost saving. To assess how the lifetime of power cables is affected by the increasing LCTs, the Arrhenius Inverse Power Model (IPM) [14] is employed to develop different scenarios of interest to estimate the average lifetime of the cable under investigation. The models and key parameters are based on the available literature/past studies [14–17]. The contribution of this paper is to combine modeling techniques with the projected loading to quantify the impact of LCTs on the lifetime of different cables. The model can be used in future studies for specific cable circuits or topologies of concern. The case studies in this paper consider an 11 kV distribution cable and 400 V mains' cable in a realistic network arrangement to assess the impact of prominent LCTs. The model results can be used to support utilities in determining the reinforcement and inspection/maintenance plans based on the projected lifetime of key cable types.

## 2. Low-Carbon Technology Adoption

There is increased focus in business and government internationally on climate change. More consumers and third parties are looking to make use of alternative technologies which are connected to the power system. This brings further challenges to the distribution utility providers around network operation, as power flows are more difficult to predict. These changes could affect cable life through increasing the loading in that portion of the network, and this may shorten life. Based on current trends, the immediate challenge is posed by electric vehicles (EVs) and heat pumps (HPs). To explore the impact on cable life, the relative uptake of these two technologies was investigated in the following subsections.

### 2.1. EV

With the cost of batteries falling rapidly and increased government incentives, it is anticipated that the stock of EVs would increase exponentially. Based on the prediction from the Global EV outlook 2021 [18], the global EV stock can grow by 13.8 times above 2020's level in 2028 and the global electricity demand from EVs would reach at least 525 TWh in 2030 (2% of global electricity total final consumption). Therefore, EVs would significantly impact power delivery and the life of assets on the distribution network. More consumers are making the switch to EVs as an environmentally friendly and cost-effective means of transport. EVs that need to be recharged can be classified as two types: plug-in hybrid electric vehicles (PHEVs) and battery electric vehicles (BEVs). A PHEV can be powered by both petrol and electricity and BEV is a pure electric vehicle. The battery of PHEVs has a lower capacity than BEVs and a reduced electric only range. A BEV normally has a DC charging port (used for rapid charging) and an AC charging port (used for destination

charging). In contrast, PHEVs normally have an AC charging port for destination charging. In the UK, in 2020, 108k BEVs were sold and 66k PHEV were sold [19]; this was 10% of the 2020 UK market share. The battery capacity of current BEVs range from 17.6 kWh in the Smart EQ ForTwo with a range of 93.3 km up to 100 kWh in the Tesla Model S and Model X, which have over 482.8 km range [20]. EV charging activity can be classified into four different typical arrangements: home, work, destination, and fast/rapid charging station. The chargers at home are typically slow (supplies 3 to 11 kW AC to the vehicle [21]), and 87% of charging activity is performed at consumers' homes [21]. Workplace charging is typically 3–11 kW AC, but charging is more likely to happen during daytime hours (approximately 9 a.m.–5 p.m.). The destination type denotes charging at shops and car parks; the chargers are generally higher power than household level but lower than rapid chargers (normally capable of supplying 7 to 22 kW) [21]. The final type is rapid chargers (can supply 22 to 120 kW—a 50 kW rapid charger could have the same impact on the network as 25 homes [21]). This enables the EV to be topped up in minutes to hours; however, such a large load interacting with the distribution network might be a challenge to manage at a large scale. DC charging speed is seen as a critical factor in the future, and charging speeds of 350 kW [22,23] will become more common as new vehicles accept this speed of charging. AC charging speed is not likely to increase much beyond the current capabilities, as there is no clear trend for increasing the rating of the on-board chargers on EVs (AC to DC converters) beyond the 7–22 kW range. This paper considers the impact of AC EV charging infrastructure at household and industrial customers.

In general, an increase in the number of EVs charging on the network would bring a higher burden for cables in distribution networks. To avoid heavy reinforcement work, an interesting option is to use a smart charging control system or incentivized energy tariffs to shift the peak of demand to off-peak periods. However, there is still a long way to go before smart charging solutions are fully implemented, and the EV stock is increasing exponentially. It can therefore be predicted that some cable runs may experience a sharp increase in the number of EVs based on local hotspots of EV uptake. This may occur before smart charging schemes are fully implemented, which may result in a significant reduction in cable life in this interim period. Even if smart charging solutions were to be fully implemented, this would not completely eliminate the impact of increased load on cables caused by EV charging. Therefore, cable life could be reduced in this period, as an increasing number of EVs are connected to the power network. As cables are the least monitored asset, modeling tools can support network operators over this interim period and beyond as further data are made available [14].

*2.2. Heat Pump*

Currently, 85% of UK households use fossil-fuel-based natural gas to heat their homes [3], making it one of the main contributors of the UK's carbon emissions. Utility providers, government, and policymakers have already conducted a range of projects [3,24,25]. The analysis [24] shows that the yearly installation rate of HP would sharply increase over the next 10 years. As an example, the yearly installation rate in 2030 would be more than 28 times the installation rate in 2020.

If conventional fossil-fuel-based natural gas boilers are replaced by HP, the electricity consumption would increase significantly. HPs come as air-source and ground-source units. The two technology types suit different installation conditions. An air-source heat pump (ASHP) is better suited to an urban environment. The ASHP absorbs heat from the air via a fluid, the fluid passes through a compressor, and this, in turn, increases the temperature of the fluid. A heat exchanger is used to transfer the heat from the fluid to the hot-water circuits of the home. A ground-source heat pump (GSHP) is more suited to a rural environment, as significant civil works are required during installation. A GSHP is more efficient, as it extracts heat from the round via buried pipework; this is termed the ground loop and has a water/antifreeze fluid inside. The system uses the same operating principal to elevate the temperature of the fluid and transfer the heat to the hot-water

circuits of the home [26]. According to References [27,28], the average power consumption ranges from 1.2 to 1.7 kW per unit, and a peak value of 4 kW was noted when one dwelling was considered. Although the demand of a single ASHP is less than a single EV, the operating time is longer, and customers are unlikely to wait until a certain time to heat their home. The flexibility options that are apparent for EV charging do not exist for ASHP, given the lack of energy storage in the power network.

An ASHP needs electricity to run, but for every 1 kWh of electricity, an ASHP can produce 3 kWh of heat [29]. The average annual thermal demand for most homes in the UK is at 12,000 kWh [29]. Therefore, this paper assumes that the average annual power consumption of the ASHP is 4000 kWh. This represents a significant increase in the average household usage (3781 kWh [30]).

## 3. Cable Lifetime Assessment Method

The previous section introduced a range of potential challenges for the most prominent LCTs. The impact will ultimately be reflected in the cables' operating temperature based on the applied load regime. This study employs a thermal aging model to estimate the cable lifetime based on the cable loading. The end-to-end workflow is given in Figure 1.

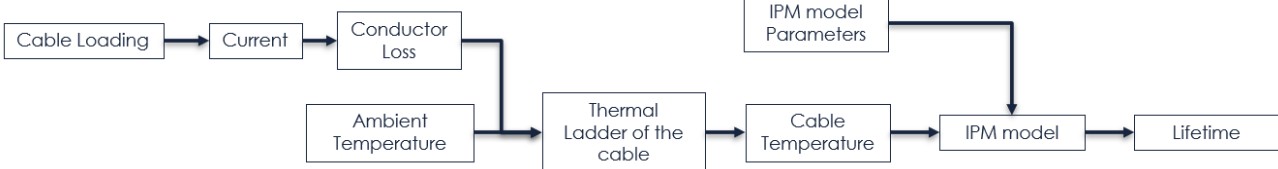

**Figure 1.** Thermal aging model for cable lifetime estimation.

As Figure 1 shows, the cable loading will increase the temperature of the conductor via the conductor losses. The conductor losses were calculated by using the conductor resistance at the rated operating temperature of the cable and the applied current due to the specified loading regime. The conductor temperature can be estimated based on the ambient temperature and the predicted temperature difference from the thermal ladder calculation. This was used as a sense check to confirm that the cable under investigation was not operating beyond the thermal capabilities of the insulation system. Finally, the Arrhenius Inverse Power Model (IPM) is used to project the lifetime with the cable temperature. To estimate the aging of a cable, the cable temperature is required. In this paper, all the heating of the cable system is assumed to come from the ambient temperature and power loss of the cable. All models were implemented in MATLAB for this study.

### 3.1. Thermal Ladder Evaluation

In this study, the transient temperature was ignored. Therefore, the cable temperature can be given as follows:

$$\theta_{cable} = \theta_{am} + \theta_{loss} \tag{1}$$

where $\theta_{am}$ was assumed to be 30 °C within this study. The temperature rise due to power losses can be simplified to a thermal ladder, which is given in Figure 2.

As Figure 2 shows, the $\theta_{loss}$ is related to $R_a$, $R_b$, and $W_c$; and $R_a$ and $R_b$ can be further decomposed into $T_A$, $T_B$, $Q_A$, and $Q_B$. Among them, $T_A$ and $T_B$ are the thermal resistance, $Q_A$ and $Q_B$ are thermal capacitances, and $W_c$ is the conductor loss. According to Reference [14], the $\theta_{loss}$ of the XLPE cable can also be written as follows:

$$\theta_{loss} = W_c \left( R_a \left(1 - e^{-at}\right) + R_b \left(1 - e^{-bt}\right) \right) \tag{2}$$

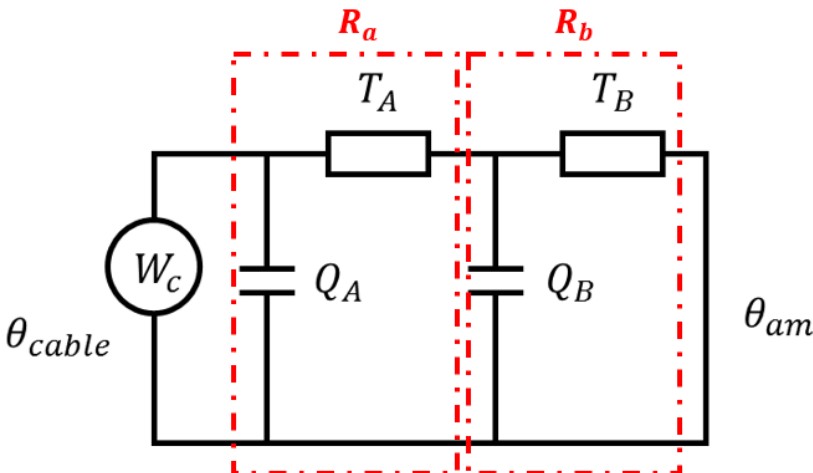

**Figure 2.** The two-loop equivalent thermal ladder representation [31].

This investigation aims to consider the average impact; therefore, $\theta_{loss}$ is assumed to be a constant value, and $t$ is assumed to be infinite. Thus, the representation of $\theta_{loss}$ can be simplified as follows:

$$\theta_{loss} = W_c(R_a + R_b) \tag{3}$$

where the variables are represented by the following [14]:

$$R_a = \frac{1}{a-b}\left(\frac{1}{Q_A} - b(T_A + T_B)\right) \tag{4}$$

$$R_b = T_A + T_B - R_a \tag{5}$$

where $a$ and $b$ are the scale parameters derived from the two-loop equivalent diagram; $T_A$, $T_B$, $Q_A$, and $Q_B$ can be further decomposed into $T_1$, $T_2$, $T_3$, $Q_1$, $Q_2$, and $Q_3$; and the calculation details can be found in Reference [14]. The thermal resistance ($T_1$, $T_2$, and $T_3$) and capacitance ($Q_1$, $Q_2$, and $Q_3$) can be calculated by using IEC 60287-2-1 [32]. Additionally, $W_c$ is the conductor loss, which can be calculated by using the following equation:

$$W_c = I^2 R * pf \tag{6}$$

where $I$ is current on the cable, $R$ is conductor resistance, and $pf$ is the power factor, which is assumed to be 1 in the study. The power factor is the ratio of active power to apparent power; hence, a power factor of 1 could indicate the worst-case conductor loss.

The thermal ladder approximation for the two cable topologies is derived in the following sections; the two topologies require specific equations from the standard [32] in the derivation of $T_1$, $T_2$, $T_3$, $T_A$, $T_B$, $Q_A$, $Q_B$, and $Wc$.

### 3.1.1. The 11 kV Distribution Cable

This section details the derivation of the thermal ladder model for the 400 mm$^2$ 3 core 11 kV XLPE cable. The key dimensions and previous modeling data were available in a past publication [33]. The specification [15] of the 400 mm$^2$ XLPE cable is given in Table 1, and the dimension is given in Figure 3.

**Table 1.** Specification of the 400 mm² 11 kV XLPE cable.

| Parameter | Value | Parameter | Value | Parameter | Value | Parameter | Value |
|---|---|---|---|---|---|---|---|
| Voltage Level (kV) | 11 | Conductor Screen Thickness (mm) | 0.7 | Sheath Diameter (mm) | - | Armor Diameter (mm) | 87.5 |
| Rated Current (Amp) | 522 | Insulation Diameter (mm) | 31.8 | Sheath Thickness (mm) | - | Armor Thickness (mm) | 3.15 |
| Conductor Area (mm²) | 400 | Insulation Thickness (mm) | 3.4 | Concentric Neutral Diameter (mm) | 35.3 | Jacket Diameter (mm) | 95.5 |
| Conductor Diameter (mm) | 23.6 | Insulation Screen Diameter (mm) | 33.6 | Concentric Neutral Thickness (mm) | 0.85 | Jacket Thickness (mm) | 4 |
| Conductor Screen Diameter (mm) | 25 | Insulation Screen Thickness (mm) | 0.9 | Armor Bedding Diameter (mm) | 81.2 | | |

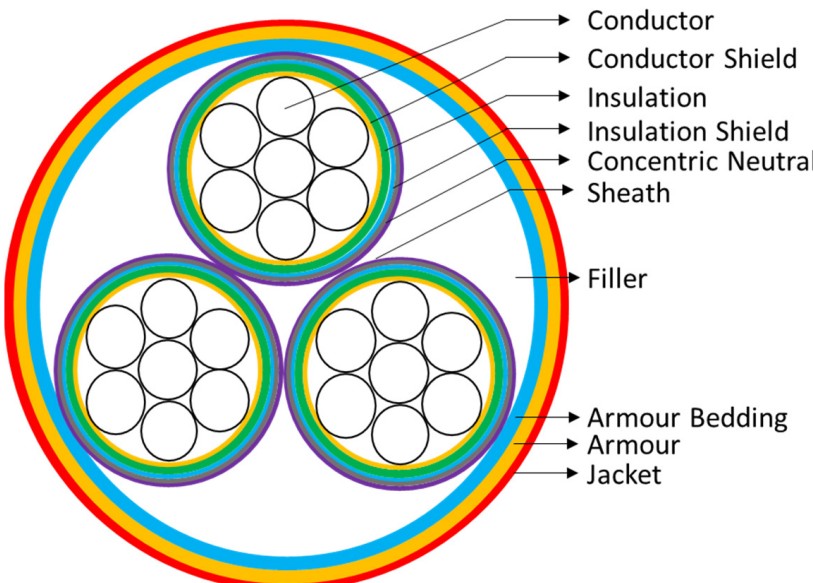

**Figure 3.** Layout of the 11 kV distribution cable.

According to IEC 60287-2-1 [32], thermal resistance, $T_1$, is the resistivity between the conductor and metallic sheath/screen; $T_2$ is the resistivity between metallic sheath and armor; $T_3$ is between armour and surroundings; and $T_4$ represents the surroundings. $T_1$ is defined below.

$$T_1 = \frac{\rho_{ins}}{2\pi}G + 0.031\left(\rho_f - \rho_{ins}\right)e^{0.67\frac{t_1}{d_c}} \tag{7}$$

where $\rho_{ins}$ and $\rho_f$ are the thermal resistivity of the insulation and the filler material respectively; $t_1$ is the thickness of the material between the conductors and outer covering; $d_c$ is the diameter of conductor; and $G$ is geometric factor, which also depends on the ratios $\frac{t_1}{d_c}$ and can be obtained from the geometric factor curve in Reference [32]. Looking up the curve, $G \approx 0.8$ can be used for the 11 kV XLPE cable in this study.

$T_2$ and $T_3$ are given by the following relationships:

$$T_2 = \frac{1}{2\pi}\rho_{ins}In\left(1 + \frac{2t_2}{D_s}\right)c \tag{8}$$

$$T_3 = \frac{1}{2\pi}\rho_{ins}In\left(1 + \frac{2t_3}{D_a'}\right) \tag{9}$$

where $t_2$ is the thickness of the bedding (in mm); $D_s$ is the external diameter of the sheath (in mm); $t_3$ is thickness of outer covering; and $D_a'$ is the external diameter of sheath, screen, and bedding.

The thermal capacitance ($Q_1, Q_2, Q_3,$ and $Q_4$) can be calculated by using the following:

$$Q = \frac{\pi}{4}\left(D_{ext}{}^2 - D_{int}{}^2\right)C \tag{10}$$

where $C$ is volumetric specific heat [33]. Based on the dimensional properties of the 11 kV XLPE cable [15], the steady-state thermal resistances and capacitors are as follows: $T_A = 0.3366$ K/W, $0.3366$ K/W, $T_B = 0.5197$ K/W, $Q_A = 1935$ J/K, $Q_B = 573$ J/K, and $W_c = 18.25$ W/m.

### 3.1.2. Mains' Cable

This section details the derivation of the thermal ladder model for the 3 core 300 mm$^2$ XLPE waveform cable. The waveform cable has sectorial shaped solid aluminium conductors. The specification and layout of the 300 mm$^2$ waveform cable are shown in Table 2 and Figure 4.

**Table 2.** Specification of the 300 mm$^2$ waveform cable [16].

| Parameter | Value | Parameter | Value |
|---|---|---|---|
| Voltage Rating (V) | 1000 | Insulation Thickness (mm) | 1.8 |
| Rated Current (Amp) | 435 | Earth Wires (mm) | 1.85 |
| Conductor Area (mm$^2$) | 300 | Sheath Thickness (mm) | 2.8 |
| Conductor Diameter (mm) | 36.8 | Overall Diameter (mm) | 55.1 |
| Rubber Bedding Layer (mm) | 0.9 | Rated Operating Temperature (°C) | 80 |

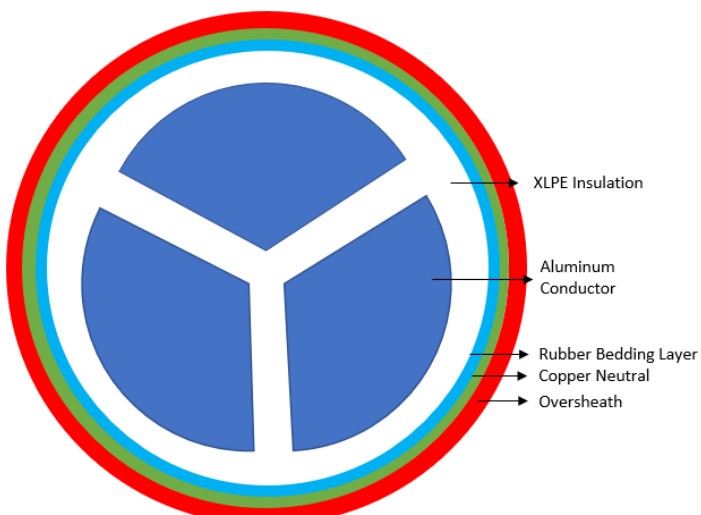

**Figure 4.** Layout of the 300 mm$^2$ waveform cable.

According to IEC 60287-2-1 [32], the waveform cable model also employed Equation (7) to calculate $T_1$. However, the geometric factor, $G$, is calculated in a different way, as outlined in Equation (12), and $F_2$ is a coefficient for the belted cable, given by the following relationships:

$$G = 3F_2 In\left(\frac{d_a}{2r_1}\right) \tag{11}$$

$$F_2 = 1 + \frac{3t}{2\pi(d_x + t) - t} \tag{12}$$

where $d_a$ is the external diameter of the belt insulation (in mm), $r_1$ is the radius of the circle circumscribing the conductor, $d_x$ is the diameter of a circular conductor, and $t$ is the insulation thickness between conductors. $T_2$ and $T_3$ are given by the following relationships:

$$T_2 = 0 \tag{13}$$

$$T_3 = \frac{1}{2\pi} \rho_{ins} In\left(1 + \frac{2t_3}{D'_a}\right) \tag{14}$$

where $t_3$ is thickness of outer covering, and $D'_a$ is the external diameter of sheath, screen, and bedding.

The thermal capacitance ($Q_1, Q_2, Q_3,$ and $Q_4$) can be calculated by using Equation (10). Using the equations above, the steady-state thermal resistances and capacitances are as follows: $T_A = 0.1798$ K/W, $T_B = 0.4840$ K/W, $Q_A = 3744.1$ J/K, $Q_B = 357.9$ J/K, and $W_c = 23.84$ W/m. The AC resistance of the 11 kV XLPE cable is 0.0645 $\Omega$/km [34]. The AC resistance of the waveform cable is 0.126 $\Omega$/km [16].

### 3.2. Lifetime Assessment with IPM

The hypothesis of cable aging is based around the IPM. The model [14] assessed how the two LCTs influenced the cable life under different scenarios. The IPM model is given below:

$$TTF = TTF_0 \frac{E}{E_0}^{-(\eta_0 - b(\frac{1}{\theta_0} - \frac{1}{\theta_{cable}}))} \exp\left(-B\left(\frac{1}{\theta_0} - \frac{1}{\theta_{cable}}\right)\right) \tag{15}$$

$$\theta_{cable} = (\theta_{am} + \varepsilon_{am}) + (\theta_{loss} + \varepsilon_{loss}) \tag{16}$$

where $E$ is electric field (kV/mm); $\theta_{cable}$ is the cable temperature in Kelvin, and it can be further decomposed into the ambient temperature $(\theta_{am} + \varepsilon_{am})$ and the joule heating in the cable conductor $(\theta_{loss} + \varepsilon_{loss})$; and $\varepsilon_{am}$ and $\varepsilon_{loss}$ represent the temperature errors. The temperature error could occur as a measurement or calculation error rather than a dynamic demand variation. This paper therefore explores the impact of $\pm5\%$ temperature error on the cable lifetime [14]. Moreover, $\theta_0$ is a reference temperature, $\eta_0$ is the voltage endurance coefficient at $\theta_{cable} = \theta_0$, $E_0$ is a value of electric field below which electrical aging is deemed as negligible (kV/mm), $TTF_0$ is time-to-failure at $\theta_{cable} = \theta_0$, and E = $E_0$. B is the ratio of $\Delta W/k$ ($\Delta W$ is the activation energy of the main thermal degradation reaction and $k$ is the Boltzman constant), and $b$ is a parameter that models the synergism between electrical and thermal stresses (K.mm/kV). The thermal rating of XLPE and PILC are at 90 °C (363.15 Kelvin) and 70 °C (343.15 Kelvin), respectively. The lifetime estimation parameters are given in Table 3.

**Table 3.** The general parameters for IPM lifetime estimation [14].

| Parameter | Value | Parameter | Value |
|---|---|---|---|
| $TTF_0$ (h) | $1 \times 10^6$ | $E_0$ (kV/mm) | 5 |
| b (K mm/kV) | 4420 | E (kV/mm) | 7.2 |
| $\eta_0$ (non-dimensional) | 15 | B (K) | 12,430 |

## 4. Case Study

This section will provide two case studies to analyze the demand impact on typical distribution network cables. Two representative cable types (11 kV XLPE cable and 400 V waveform cable) are considered in this study and are introduced in the following sections.

### 4.1. 11 kV Distribution Network

For the 11 kV distribution cable, the example network has 24 distribution transformers which step down to 400 V, and there are a total of 5032 customers [17]. Among these

customers, approximately 5.3% are commercial and the remainder are domestic. According to a 2018 report from the Department for Business, Energy, and Industry Strategy [30], the mean annual domestic electricity consumption per meter in GB in 2016 was 3781 kWh. The mean annual industrial/commercial electricity consumption in 2016 was 68,460 kWh. The example network has an average loading of 4.14 MW. By employing the workflow presented in Figure 1, the lifetime projections for different loading regimes can be derived, and the impact of temperature error is also considered. The variation of cable lifetime with applied current is outlined in Figure 5.

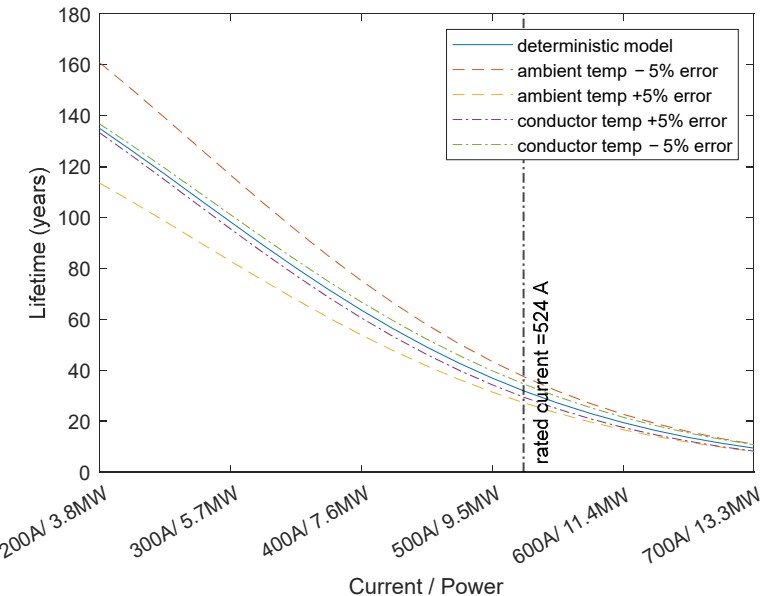

**Figure 5.** The lifetime curve for applied current and loading for the 11 kV XLPE cable.

As Figure 5 shows, if the rated current (for 10 MW rated power) is applied throughout the cable operation, a lifetime of 31.7 years is projected with the deterministic model. A 5% ambient temperature error could result in a ~15% variation lifetime, and a 5% conductor temperature error could incur an 8% variation on lifetime estimate. Ambient temperature error had a more significant impact on the cable lifetime estimation when the cable loading was low. Based on the analysis above, the 5% error in ambient temperature has a larger impact on the lifetime prediction. Generally, heavier cable loading would exceed the operating temperature of the cable and, thus, would significantly reduce the cable life or cause premature failure. The lifetime of the cable with a constant load of 4.14 MW is projected as 129 years ± 22 years. The use of mean cable loading does not account for the peak-times in the load profile. Therefore, this could lead to an underestimate of lifetime. Based on this example, the 11 kV cable is operating at 50% of its rated capacity in the normal baseline scenario. This will not be the case for all portions of the network, and the ability to pinpoint these issues before they occur will be paramount to future network operation. Some variations are presented based on the impact of EVs and ASHPs in the following sections.

### 4.2. Mains' Cable Network

The mains' cable is assumed to supply 200 domestic customers from the local distribution transformer. The example network has an average loading of 86.32 kW. This level of loading, on average, would equate to a lifetime of 65 years. The following sections discuss the impact of EVs and ASHPs on cable lifetime. The following calculations assume that the operating voltage for the cable is 400 V. The impacts of applied current and loading on the lifetime are given in Figure 6.

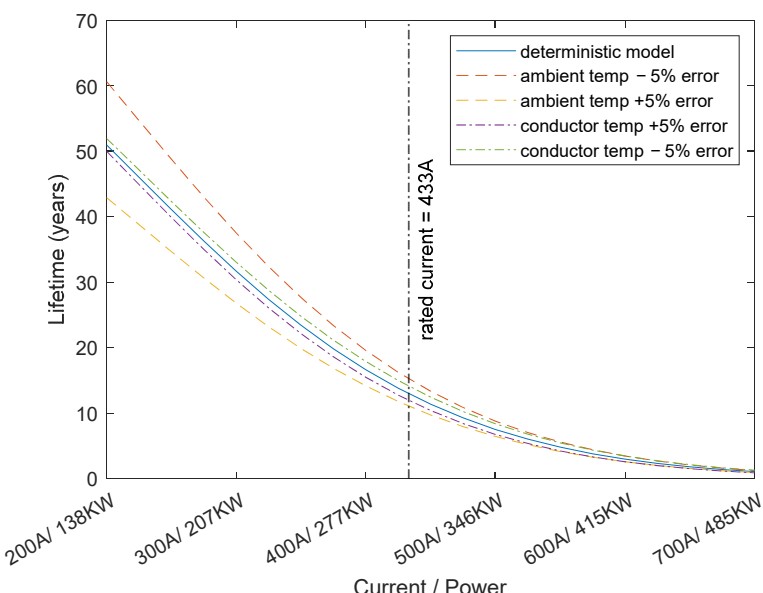

**Figure 6.** The lifetime curve for applied current and loading for the 300 mm² waveform cable.

As Figure 6 shows, the loading will determine the lifetime of the mains' cable, and when operating at the rated current, the lifetime drops significantly to 12.8 years (deterministic model value). The conductor temperature and ambient temperature error result in ±8% and ±15% lifetime variation, respectively. To investigate how the technologies influence the cable life, investigations were based on an example network.

*4.3. EV Impact*

This section investigates the impact of EV charging on the example networks introduced above. According to the background review in Section 2.1, each domestic charger consumes around 3 to 7 kW, and an industrial/commercial charger could be from 7 to 22 kW. This investigation assumes that each household has 6 h for charging every day; this may be an over estimation in itself. The EV loading is assumed to be an additional load at each domestic/non-domestic property.

The new average cable loading, $L_{new}$, can be calculated with the following equation:

$$L_{new} = L + (p_d \times r_d \times n_d + p_i \times r_i \times n_i) \times w \tag{17}$$

where $L$ is the conventional cable loading; $p_d$ and $p_i$ are the average power consumption of domestic and industrial customers, respectively; $r_d$ and $r_i$ are the usage rate of the LCT technologies for domestic and industrial customers, respectively; $n_d$ and $n_i$ are the number of domestic and industrial customers on the operational cable, respectively; and $w$ is the adoption rate of EVs.

4.3.1. The 11 kV Distribution Cable

The EV loading on the 11 kV distribution cable was considered first. The best (lowest possible average consumption increase), median, and worst scenarios (largest increase) are given in Table 4.

In all scenarios, the introduction of EVs would increase the loading of the cable. This would significantly reduce the lifetime of the cable. With an adoption rate of 100% in the best-case scenario, the cable lifetime would drop down to 54 years. In the worst-case scenario, the cable would fail in 7.3 years if the rated conditions of the cable were ignored. In reality, the adoption rate would not be 100% in the near future. Figure 7 outlines the effect on penetration rate for the three scenarios.

**Table 4.** Three different scenarios for EV consumption at 11 kV.

| Scenario | Average Consumption Increase | | Demand at 100% Adoption Rate |
| --- | --- | --- | --- |
| | **Household Customer** | **Industrial Customer** | |
| Best-Case Scenario | 3 kW | 7 kW | 8.18 MW |
| Median Scenario | 7 kW | 11 kW | 13.21 MW |
| Worst-Case Scenario | 7 kW | 22 kW | 13.94 MW |

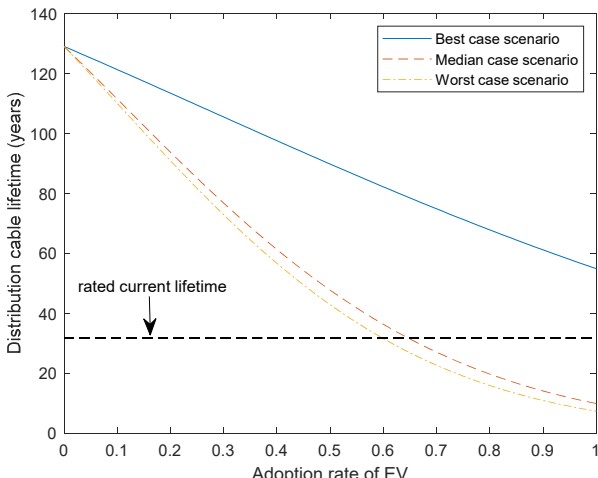

**Figure 7.** The lifetime of the distribution cable with EV adoption rate.

4.3.2. Mains' Cable

Two scenarios are used to analyze the EV impact on the LV mains' cable; only domestic customers are considered in this case. If each household has an EV, the cable loading would be given by two different assumed load levels (3 and 7 kW). In this case, the new load equation is expressed as follows:

$$L_{new} = L + (p_d \times r_d \times n_d) \times w \tag{18}$$

As Figure 8 shows, the presence of EVs can significantly reduce the lifetime of a mains' cable. For the worst-case scenario, the lifetime can be reduced to approximately 2.5 years if the rated conditions are exceeded. An EV adoption rate of 60% in the worst-case scenario represents the threshold at which the rated conditions of the cable have been exceeded.

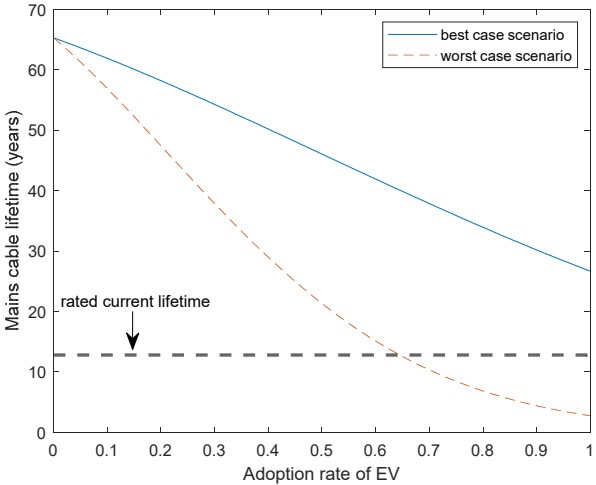

**Figure 8.** The lifetime of the mains' cable with EV adoption rate.

### 4.4. Heat-Pump Impact

As discussed in Section 2.2, the average power consumption of an ASHP is assumed as 4000 kWh per year. Based on government incentives/goals, the adoption rate of ASHP is expected to rise, and the load on distribution cables will rise sharply. This work assumes that the domestic consumption for a single household is the same as that of an industrial customer. The impact on cable life with ASHP adoption is shown in Figure 9.

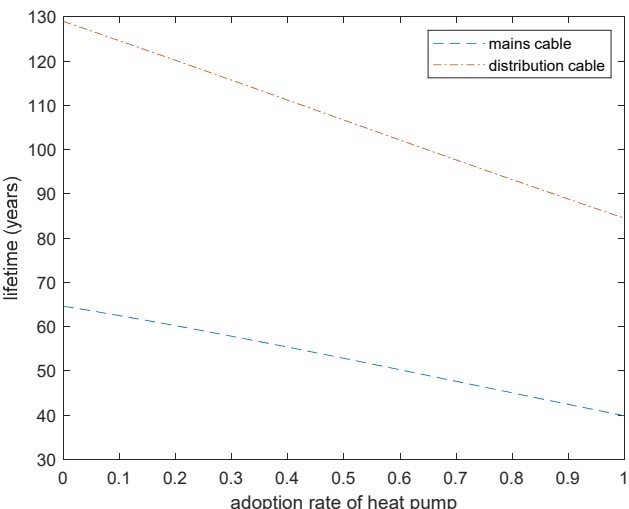

**Figure 9.** Impact of adoption rate of ASHP on the life of the mains' cable.

If every household on the example distribution network installed an ASHP (adoption rate of 1), the total demand would be 6.44 MW, and the cable life would be reduced to approximately 84 years from a base case of 127 years with no ASHP penetration. The lifetime of the distribution cable generally drops faster than the mains' cable.

The mains' cable is assumed to have only domestic customers. In the example network, when conventional boilers are replaced with ASHPs, the lifetime estimation for the mains' cable decreases sharply. Based on the 200-customer example network, when the adoption rate reaches 100% (cable loading is 177.6 kW), life would decrease to around 40 years. In the example network, this level of penetration could be permitted without exceeding (on average and with ASHPs the only contributor to a customer load increase) the rating of the mains' cable.

## 5. Impact of Mixed LCT

As discussed in the previous section, the growth of EVs and ASHPs could have a significant impact on the life of cables in the near future. This section uses the projections for EV and ASHP uptake combined with the IPM model to estimate how *LCT* loading could impact the end-life of cables. This analysis will employ the network examples in Section 4. According to Reference [33], the number of households in the UK is around 27.8 million. The adoption rate ($Rate_{LCT}$) of EVs and ASHPs are considered separately and employs the following relation:

$$Rate_{LCT} = \frac{S_{LCT}}{N_H \times (1 + g_r)^n} \tag{19}$$

where $S_{LCT}$ is the stock of the *LCT*, $N_H$ is the number of households under consideration, $g_r$ is the growth rate, and n is the number of years under consideration. Additionally, this analysis assumes that the number of UK households will grow by 0.9% every year [35] and that the EV stock growth rate follows the global estimate. Based on Reference [36], in the UK, there were approximately 450,000 EVs in 2020. Therefore, the EV adoption rate in 2020 was calculated as 1.6%. The global EV stock was 10.2 million in 2020 and is projected to be

141.1 million in 2028 [18]. Based on this projection, the UK EV adoption rate in 2028 would be 20.66%

According to Reference [24], the UK HP installation rate in 2020 was projected as 36,000, and the UK HP installation rate in 2028 is projected at 714,000. According to Reference [37], the installed capacity of HPs was 238,823 by 2019. This analysis also assumes that the household size in 2019 was 27.8 million and the growth rate is constant at 0.9%. Therefore, the HP adoption rate in 2020 was calculated as 0.98%, and the adoption rate in 2028 was calculated as 9.52%.

This study assumes that the mean cable lifetime is based on the loading regime projected by the best/worst-case scenarios. The combined effect from EVs and HPs is based on the anticipated adoption rates discussed above. The projected values are given in Tables 5–8. Based on Figure 5, the largest change in the lifetime prediction was observed for variations in ambient temperature. Only the effect of ambient temperature error is considered in Table 8.

**Table 5.** Adoption rate of EVs and HPs in 2020 and 2028.

| 2020 | | 2028 | |
|---|---|---|---|
| **Rate$_{EV}$** | **Rate$_{HP}$** | **Rate$_{EV}$** | **Rate$_{HP}$** |
| 1.6% | 0.98% | 20.66% | 9.52% |

**Table 6.** Distribution cable loading for 2020 and 2028.

| Property | | 2020 | | 2028 | |
|---|---|---|---|---|---|
| | | EV | HP | EV | HP |
| **Rate** | | 1.6% | 0.98% | 20.66% | 9.52% |
| **Consumption** | **Best case** | 64.5 kW | 21.1 kW | 0.8333 MW | 0.2187 MW |
| | **Worst case** | 156.6 kW | | 2.022 MW | |
| **Total** | **Best case** | 85.6 kW | | 1.052 MW | |
| **Consumption** | **Worst case** | 177.7 kW | | 2.241 MW | |

**Table 7.** Mains' cable loading for 2020 and 2028.

| Property | | 2020 | | 2028 | |
|---|---|---|---|---|---|
| | | EV | HP | EV | HP |
| **Rate** | | 1.6% | 0.98% | 20.66% | 9.52% |
| **Consumption** | **Best case** | 2.4 kW | 0.895 kW | 30.99 kW | 8.69 kW |
| | **Worst case** | 5.6 kW | | 72.31 kW | |
| **Total** | **Best case** | 3.295 kW | | 39.68 kW | |
| **Consumption** | **Worst case** | 6.495 kW | | 81 kW | |

**Table 8.** Lifetime prediction changes (years) for 2020 and 2028, considering ±5% ambient temperature error.

| Property | Projected Lifetime (Years) | | | |
|---|---|---|---|---|
| | Distribution Cable | | Mains' Cable | |
| | **Best Case** | **Worst Case** | **Best Case** | **Worst Case** |
| **2020** | 127 ± 20 | 125 ± 20 | 64 ± 10 | 63 ± 10 |
| **2028** | 108 ± 18 | 85 ± 13 | 54 ± 9 | 42 ± 7 |
| **Difference** | −19 | −40 | −10 | −21 |

The scenarios in 2020 have a limited impact on the cable lifetime for both distribution and mains' cables. As the adoption rate increases, as projected in 2028, the impact becomes

more significant. When compared to the 2020 best-case scenario, the average lifetime of both distribution and mains' cable could reduce by up to 30% (it could be up to 67% out if the worst-case error is considered) for the projected worst-case scenario in 2028. This estimation is based on the average load; the actual lifetime may reduce more significantly if the peak loading is considered. Generally, the rapid uptake of both electric vehicle and heat pump adoption in the next eight years will have a significant impact on cable life, and a large number of cables would need to be upgraded. The proposed model with more specific network data can be used to strategically target replacement and reinforcement, which will significantly save costs for utilities, whilst still maintaining standards of service.

## 6. Conclusions

The targets for net zero around the world have placed further emphasis on the adoption of LCTs in domestic and industrial electricity customers. This increased adoption of LCT can bring further loading for the power network as a whole. The demand of both customer profiles is projected to increase, and this presents a challenge for network operators to ensure that the network will cope with this changing need. The widespread deployment of sensors and measurement systems is not practical or cost effective on the LV distribution network. This paper has demonstrated a temperature-based end-of-life estimation model to derive a relation between cable loading and lifetime. Two representative cable topologies and associated example networks were employed to explore the challenge. The study found that EVs and HPs are most likely to be the prominent technologies adopted in the short-to-longer term. Both technologies increase the cable loading and could reduce the cable life by up to 30% by 2028, based on the projected uptake rate of EVs and HPs.

In general, EVs would have a significant impact on the customer demand and cable lifetime. The scenarios mentioned above are very simplified; however, this can be developed further to be more realistic for specific network scenarios. An additional complication may be the development of vehicle-to-grid chargers; trials are ongoing in this area, so EVs may become a source, as well as a load. A network of EVs working as sources could be a potential future mitigation method to reduce the overall power imported to portions of network if constraints exist.

HPs alone might not cause cable overload problems, but the combination of this load increase along with other LCTs may pose challenges to the power system. It is anticipated that the initial uptake of HPs will be highly dependent on the prior source of heating to the customer. It is likely that government incentives will target the most polluting heating systems first. Furthermore, some customers will not be on the gas mains network and will be reliant on oil heating systems. This initial uptake may lead to localized pressures on the distribution network in areas where no gas mains are laid/available for customers to be connected.

The calculations within this paper are based on the average load increase at the two customer types, and this may present an underestimate of the challenge facing network operators. This work represents an initial step in the use of modeling tools to support the end-of-life assessment of cables, whilst further monitoring tools are deployed on the power network. Advantages of this approach center on the ability to perform long-term assessments for installed cable assets. This can enable the study of a range of possible future scenarios and varied installation conditions. The disadvantages of the approach may involve the quality of models or input data, with any errors at these stages impacting the accuracy of the lifetime prediction. Future work will consider the temperature impact on the AC conductor resistance, additional cable topologies, time-based behavior of the identified loads/sources, and potentially the uptake of DC EV chargers. The ultimate aim is to provide a tool for asset managers to make informed investment decisions and reduce the reliance of reactive maintenance/replacement schemes.

**Author Contributions:** X.J., methodology, software, validation, formal analysis, investigation, data curation, writing—original draft preparation, visualization, and writing—review and editing; E.C., conceptualization, resources, writing—review and editing, visualization, validation, supervision, project administration, and funding acquisition; B.G.S., conceptualization, methodology, writing—review and editing, validation, supervision, and funding acquisition; B.S., conceptualization, methodology, writing—review and editing, validation, supervision, and funding acquisition. All authors have read and agreed to the published version of the manuscript.

**Funding:** This work was funded by the industrial members of the Power Network Demonstration Centre (PNDC) through the Core Research Programme, and this support is gratefully acknowledged.

**Institutional Review Board Statement:** Not applicable.

**Informed Consent Statement:** Not applicable.

**Data Availability Statement:** Not applicable.

**Acknowledgments:** The technical support and input from industrial partners through the lifecycle of the project is gratefully acknowledged.

**Conflicts of Interest:** The authors declare no conflict of interest.

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
