# Peer review of "Impact of Increased Penetration of Low-Carbon Technologies on Cable Lifetime Estimations"

_electricity, doi:10.3390/electricity3020013_

Round 1

Reviewer 1 Report

The subject is very relevant. A have a few questions/remak, page 3, the 350 kW estimate is a high number, please include a reference to this and comment wether that can be introduced without a significant change of cable dimensions. On page 4, Figure 1 shows that the conductor loss is determined before temperature, as temperature influence the resistance of the cable conductors and therefore also the magnitude of the losses which will influence the final temperature, how is that taken into consideration when performing the calculations in the rest of the paper? Please clarify.

How the cable is placed (ground or on cable ladder etc) will influence the result, can the influence of this be discussed in the discussion part?

Minor comments: page 9 remove/fix the "error message". Page 10 do not end the page with a headline. Page 11 in the table change "w" to "W" in the table. Check through the text that there is space between the number and the unit (not 177.6kW but 177.6 kW". Figures 7-9 will be hard to read without color, please use different line thicknesses or pattern.

Author Response

See attached file detailing our responses.

Reviewer 2 Report

Dear Authors,

Thank you for the submission to this Journal. My comments are reported in the attached PDF.

Regards

Author Response

See attached document itemising comments and detailing changes resulting from our responses.

Reviewer 3 Report

In the reviewed manuscript a temperature based end of life estimation model to derive a relation between cable loading and lifetime was demonstrated. Two representative  cable topologies and associated example networks were employed to explore the challenge. In my opinion the reviewed manuscript is complete and I would like to see this paper publish in Electricity Journal.

Apart from minor editorial errors (e.g. in line 296), the article can be supplemented with further options for modification/extension presented researches. Moreover, in section Conclusion, should be mentioned advantages and disadventages of the presented researches.

Author Response

Please see attached file for our response to the reviewers comments

Round 2

Reviewer 2 Report

Thank you for answering and addressing the comments from the first stage of revision.

Regards.